# Psychometric Properties of the Colombian Version of the HIV Attitudes Scale for Adolescents

**DOI:** 10.3390/ijerph17134686

**Published:** 2020-06-29

**Authors:** Mayra Gómez-Lugo, Alexandra Morales, Alejandro Saavedra-Roa, Janivys Niebles-Charris, Paola García-Roncallo, Laurent Marchal-Bertrand, José Pedro Espada, Pablo Vallejo-Medina

**Affiliations:** 1SexLab KL, School of Psychology, Fundación Universitaria Konrad Lorenz, Bogotá 110231, Colombia; mayraa.gomezl@konradlorenz.edu.co (M.G.-L.); diegoa.saavedrar@konradlorenz.edu.co (A.S.-R.); laurent.marchalb@konradlorenz.edu.co (L.M.-B.); pablo.vallejom@konradlorenz.edu.co (P.V.-M.); 2Grupo Aitana, Departament of Health Psychology, Universidad Miguel Hernandez, 03202 Alicante, Spain; jespada@umh.es; 3Social Sciences Department, Universidad de la Costa, Barranquilla 080002, Colombia; jniebles4@cuc.co (J.N.-C.); pgarcia5@cuc.edu.co (P.G.-R.)

**Keywords:** HIV/AIDS, attitudes, validity, adolescent, Colombia

## Abstract

The HIV Attitudes Scale (HIV-AS) evaluates attitudes towards different aspects of HIV. In view of the lack of scales measuring this construct in Colombia, this study sought to validate the HIV-AS test for adolescents from Colombia. A total of 867 Colombian students, aged between 14 and 19 years (M = 15.97 years; SD = 1.37) were evaluated. Participants responded to the HIV-AS test and a set of scales used to assess external validity. Content validity analyses reflected good adequacy indices for the items. Exploratory factor analyses revealed a four-factor structure and reliability indices were satisfactory. The structural equation model showed good levels of fit. Most of the items presented a discrimination index above 0.30 and contributed to the reliability of the scale, except for item 9, which was eliminated. Concurrent validity showed significant correlations among the HIV-AS and other similar constructs. A reliable measurement of attitudes toward HIV allows for an improved assessment of the risk associated with exposure to sexually transmitted infections in adolescent populations.

## 1. Introduction

The current report by the Institute for Health Metrics and Evaluation states that at least 37.9 million people worldwide are infected with the HIV virus [1]. Moreover, 32.0 million people have died from AIDS-related illnesses since the start of the epidemic [2]. With respect to pediatric and adolescent populations, The Joint United Nations Programme on HIV/AIDS (UNAIDS) estimates more than 1.7 million cases of sexually transmitted infections (STIs) among children under 15 years of age; 42% of STIs affect young people between 15 and 24 years of age [3]. Moreover, AIDS is currently the second most frequent cause of death among people between 10 and 19 years of age around the world; most of these deaths are due to lack of treatment because the serological status of these young people is unknown [3].

In the global scenario, Latin America and the Caribbean stand out as the areas of highest risk of new infections. Data estimates from 2018 state that there are 1.3 million people carrying the HIV virus in these countries, one third of which are adolescents and young people between 15 and 24 years of age [4]. In 2019, Colombia had one of the highest virus transmission rates in Latin America, with close to 0.4% of its population infected [1]. Consequently, the HIV epidemic in Colombia has been tagged as ‘focused’, which means that the HIV virus has rapidly spread among one or more populations. Although it has not yet spread among the general population, this figures a pressing need of attention [1]. Therefore, prevention, treatment, attention, and support efforts should be focused on key high-risk populations, among them the adolescent population.

Various theoretical approaches have been developed in order to understand and modify sexual risk behaviors among adolescents. For instance, Ajzen’s [5] Theory of Planned Behavior (TPB) has shown a higher predictive power in regard to consistent use of condoms than other social-cognitive approaches [6,7]; an individual’s intention of adopting a behavior is always influenced by three factors: the individual’s attitude toward the behavior, the influence of social factors, and the individual’s perception of behavioral control. Applied to the study of HIV, TPB focuses, among other elements, on attitudes defined as a learned tendency to respond in a certain way to a specific situation, which becomes a main component in the establishment of healthy sexual behaviors [8]. According to systematic reviews [9,10], attitudes are indeed the construct most closely associated with healthy sexual behaviors, such as consistent use of condoms in sexual relationships or taking an HIV test, which are fundamental in developing a sexual health promotion program.

Given the importance of evaluating attitudes to different aspects of HIV/AIDS, several scales are currently available, for example the EA-AIDS HIV/AIDS Attitude Scale, validated for Portuguese-speaking populations [11]; the Stigmatizing Attitudes Towards People Living With HIV/AIDS Scale (SAT-PLWHA-S) [12], validated for Quebec; the HIV/AIDS-164 Scale [13], validated for adult Colombian [14], and its short version, the HIV/AIDS-65 Scale [15], validated for adult Spanish population [16]; the scale designed by Boileau, Rashed, Sylla, and Zunzunegui [17] for adolescents and young people from West Africa; or the HIV Attitudes Scale (HIV-AS) [18], validated for Spanish and Portuguese adolescents populations [19]. The last scale in this list, the HIV-AS, is currently the only scale designed for Spanish-speaking populations to evaluate attitudes associated with HIV/AIDS exclusively among adolescents. It is a multidimensional scale that uses a small number of items to evaluate different attitudes relevant to sexual health. The HIV-AS has satisfactory psychometric characteristics and is it easy to understand and administer. Therefore, given the need of an instrument meeting such criteria that could be used with Colombian adolescent populations, the objective of the present instrumental study was to carry out a cultural adaptation and to evaluate the reliability and validity of the HIV-AS scale among Colombian adolescents [20].

## 2. Materials and Methods

### 2.1. Participants

The sampling procedure was based on a non-probabilistic incidental method. It consisted in evaluating 867 students from 12 educational centers (458 women and 396 men) living in the cities of Bogotá (467) and Barranquilla (400), aged between 14 and 19 (M = 15.97 years; SD = 1.37); sociodemographic data are presented in Table 1.

### 2.2. Instruments

Sociodemographic questionnaire. Different sociodemographic characteristics of the participants were measured using an ad-hoc semi-structured survey. Surveyors asked about age, sex, educational level, religion, religiousness, socioeconomic level, family situation, and whether the subject had a partner. Additionally, the survey included questions about consistent use of condoms, self-efficacy about condom use, peers’ perceptions about using condoms, and sexual orientation (using Kinsey’s scale).

Scale of attitudes toward aspects of HIV for adolescents (HIV-AS; [18]). The instrument consists of 12 items, each with four response alternatives (from 1 = Completely disagree to 4 = Completely agree) distributed among four factors; the first factor evaluates obstacles to safe sex, the second evaluates attitudes toward the HIV test, the third explores attitudes toward using condoms, and the fourth factor examines attitudes toward people living with HIV. The internal consistency of the questionnaire ranged between 0.56 and 0.73 in the present study. The questionnaire is presented below as an Appendix A (Table A1).

Multicomponent AIDS Phobia Scale (MAPS; [21]), version validated for Colombia [22]. This scale evaluates phobic attitudes toward AIDS by focusing on the main dimensions of the disease. It consists of 20 Likert-type six-point response items where 0 represents complete agreement and 5 represents complete disagreement. This study used the Colombian Spanish version adapted by Vallejo-Medina et al. (2018). The scale uses a two-factor structure; a sample item for fear of the disease (α = 0.67) is “I am afraid that I will die from AIDS,” and a sample item for fear or avoidance of people with AIDS (α = 0.77) is “I would feel comfortable in a room with a friend who had AIDS.”

Short Health Anxiety Inventory (SHAI; [23]), version validated for Colombia [24]. This self-administered test is used to evaluate concern about health, bodily sensations, and fear of the negative consequences of suffering from a disease. It consists of 18 items, each with four response options scored with 0 (lack of symptoms), 1 (mild symptoms), 2 (severe symptoms), and 3 (very severe symptoms). A sample item is: “Thoughts about being sick are so strong that I don’t even try to resist them anymore.” The SHAI test has shown acceptable psychometric properties when tested on clinical and non-clinical populations. The internal consistency of the Colombian version of the scale for the Fear of the disease factor is α = 0.80, and its internal consistency for the negative consequences of the disease factor is α = 0.68.

Knowledge about HIV and other Sexually Transmitted Diseases Scale (KSI; [25]) Colombian version [26]. This scale is composed of 24 items, grouped in five subscales: (a) Overall HIV knowledge about HIV; (b) Condom knowledge; (c) HIV transmission knowledge; (d) Knowledge about other STIs and (e) HIV prevention knowledge. The internal consistency of the Colombian version is 0.87. A sample item is “HIV affects the human immune system.”

### 2.3. Procedure

The cultural adaptation of the HIV-AS test to Colombia was conducted using the guidelines provided by Vallejo-Medina [27], which recommend translating and adapting the scales to different cultural contexts. Firstly, four psychologists with at least one postgraduate degree adapted the Castilian Spanish questionnaire to the Spanish used in Colombia’s cultural context. These four experts met the criteria of being Colombian natives living in the country, as well as having studied at least one year in Europe. Once the items were adapted, two expert sexologists, along with the four psychologists, verified whether the items had been correctly adapted per guidelines by Muñiz, Elosua, and Hambleton [28].

Once consensus on the adapted version was achieved, another four Colombian psychologists working in the area of sexuality and/or psychometrics evaluated the properties of the HIV-AS adapted version. The following properties were evaluated: representativeness, contribution of item to construct (attitudes toward HIV); pertinence, item belongs within a given factor (obstacles, HIV test, use of condoms, and people living with HIV/AIDS); comprehension: whether the item is understandable in its adapted version; interpretation, evaluates the item’s level of ambiguity; and clarity, how concise or straightforward the item is. The experts assessed each item with respect to each of these properties and scored the items using a Likert-type scale from 1 (Not at all) to 4 (very). Additionally, the experts were able to propose an alternative item wording in case they deemed it necessary. The next step consisted in calculating the expert agreement percentage; all items in which recommendations by experts were below 80% of agreement were revised and changed if required. As previously stated, the main sampling process, of an incidental type, was performed at 12 educational centers in the Colombian cities of Bogotá and Barranquilla. The questionnaire was administered simultaneously to all students in school classrooms using a written form. Surveyors were professional psychologists who had been trained prior to administering the questionnaire.

### 2.4. Ethical Considerations

Project execution guaranteed compliance with national, institutional, and international regulations concerning participants’ well-being. Informed consent forms were submitted to participants and participants’ parents before the questionnaire was administered to register authorization to participate in the study. As previously stated, questionnaires were administered collectively in classrooms by trained surveyors. Participants were informed that they were free to leave the study at any time and were assigned an identification code to guarantee their anonymity. They were also given the opportunity to request information about the study if they desired it.

### 2.5. Statistical Analysis

Qualitative analysis used the item specification table [29] and the ICaiken software (San Martín de Porres University and Federico Villarreal National University, Lima, Perú) [30], which allowed for the identification of Aiken’s V confidence interval [31]. All items with scores lower than 0.50 in Aiken’s V lower interval (CI = 95%) were rejected [30]. Comments shared by experts were considered to make relevant adjustments. 

The psychometric properties of the items were assessed using SPSS statistical software (version 20.0). Exploratory factor analysis (EFA) was carried out using FACTOR software (version 9.3.1) (Universitat Rovira i Virgili, Tarragona, Spain) [32]) on a randomly selected sub-sample (*n* = 350). The polychoric correlation matrix was used in this part of the analysis, which is adequate for ordinal scales [33]. Additionally, a parallel analysis was conducted as an extraction method because this procedure has demonstrated higher precision to obtain the number of factors than traditional extraction criteria [34]. The reliability of each scale was obtained by ordinal alpha, a less biased indicator for categorical response scales than Cronbach’s alpha [35]. Finally, EQS software (version 6.1) was used to carry out a confirmatory factor analysis on a random sub-sample (*n* = 511) obtained by maximum likelihood robust (ML-R) estimation. The polychoric matrix was also used for this analysis.

## 3. Results

### 3.1. Item Analysis

Table 2 presents a qualitative evaluation of the items developed by sexuality and psychometrics experts. As can be observed, Aiken’s V 95% lower limit was always higher than 0.50, which suggests the adequacy and functionality of the items.

### 3.2. Exploratory Factor Analysis

The Kaiser-Meyer-Olkin (KMO) coefficient was 0.77 and the sphericity test was significant at a *p* < 0.001 level, which indicates that data were appropriate to conduct factor analysis.

As in the original version, four factors explained 70% of the variance (Table 3). Factor 1, obstacles, uses three items to evaluate attitudes toward using condoms in situations when healthy behaviors can be thwarted by negative influence from other people or by barriers to using condoms. Factor 2, attitudes toward the HIV test, employs two items to evaluate situations in which people take or recommend taking the test when risk of infection is involved. Factor 3, use of condoms, uses four items to measure the subject’s inclination to defend the use of condoms and to have condoms ready for use. Factor 4, people living with HIV, explores attitudes toward people who carry the HIV virus taking into account contexts in which the seropositive person is close to the survey respondent; initially, this factor included three items, but the elimination of one of these items increased the explained variance percentage from 66% to 70%.

### 3.3. Item Reliability and Psychometric Properties

Once the Colombian version of the HIV-AS was found to have a four-factor structure, as the original version, some of the items’ psychometric properties were examined by calculating mean, standard deviation, corrected item-total correlation, ordinal alpha when item is removed, and subscale reliability. Table 4 shows the general adequacy of our version reflected by the indicators; however, an increase in ordinal alpha (from 0.57 to 0.62) when item 9 was removed suggested reliability problems in Factor 4: people living with HIV. Moreover, corrected item-total correlations (rit ^c^) were higher than 0.30, a value deemed as satisfactory [36], for all items except item 9, whose value was 0.24. When item 9 was removed, the values of items 8 and 10 increased from 0.38 and 0.42, respectively, to 0.44 in both cases. On the other hand, item mean values were closer to the theoretical response mean (2.5), and standard deviations were close to 1.

### 3.4. Confirmatory Factor Analysis

Confirmatory factor analysis showed satisfactory adequacy indices as compared with the original four-factor model S-Bχ2 (38) = 112.030, *p* < 0.001, confirmatory fit index (CFI) = 0.96, and root mean square error of approximation (RMSEA) = 0.062 (CI 90% RMSEA = 0.049–0.075). Figure 1 presents a diagram of the standardized results of the model, associated factor weights (λ), errors in each item, and item variance (explained for each factor), as well as covariances among factors.

### 3.5. Concurrent Validity

Concurrent validity was analyzed using Pearson’s correlations among the dimensions of obstacles, HIV test, condom use, people living with HIV, and MAPS, SHAI, and KSI subscales. In general, the data show slight to moderate positive and negative correlations. Table 5 presents magnitude and direction of each correlation.

## 4. Discussion

Attitudes toward HIV are an important construct in the efforts to reduce risky sexual behaviors [9]. Despite the significance of the construct, few instruments allow for a brief and thorough evaluation. Therefore, the present study adapted the HIV-AS test to the Colombian culture and assessed the structure of its factors and psychometric properties when used with a sample of Colombian adolescents. Results show that our adaptation of the scale is valid for Colombia. 

A qualitative analysis showed that the items were adequately constructed: Aiken’s V 95% limit was higher than 0.50 in all cases. The exploratory factor analysis revealed a four-factor structure that reflects the original version and the Portuguese version [17]; all of the factors in our version explain 70% of total variance, whereas in the original version factors explain 64,63% of total variance [19]. In general, the indicators of the psychometric properties of the items were satisfactory. However, item 9 (Factor 4) presented issues, since it decreased the percentage of variance explained by the factors, its corrected item-total correlation was lower than 0.30, and it penalized the reliability of the scale. Therefore, the research team decided to eliminate the item (“Would you kiss a person in the cheek who had HIV?”) for the Colombian version. On the other hand, as recommended by Carretero-Dios and Pérez [20], the mean scores of the questionnaire items are very close to the theoretical mean, and its standard deviations are close to 1. 

The external validity coefficients of the HIV-AS subscales were very similar to those found in other studies [18]. The dimensions of people living with HIV, fear of other people, and fear of infection of the MAPS test carried significant moderate negative correlations with one another, which can be explained in the sense that attitudes toward HIV are the base components of AIDS phobia [37]; the same correlation took place between the dimensions of obstacles and fear of others/avoidance in the MAPS test, reflecting findings from other studies from Spain [38]. On the other hand, the dimensions of people living with AIDS and condom use had moderate negative correlations with the dimension of fear of negative consequences in the SHAI test. In the first case, the correlation matches findings by Lau and Tsui [39], whose investigation revealed a relationship between the perception of risks associated with the disease and its consequences and attitudes towards people living with AIDS. In the second case, results are in line with a study by Uribe, Orcasita, and Vélez [40] with Colombian population, who found a relationship between susceptibility to the consequences of diseases and attitudes toward health protection behaviors, such as using condoms. Moreover, moderate positive correlations were found among all the HIV-AS subscales and most of the factors in the scale of knowledge about HIV and other STDs, which suggests that more knowledge results in better attitudes toward HIV, as has been found by theorical authors as Ajzen [5] and authors working with the Portuguese version of the scale [19] and also by authors studying Colombian populations [41,42]. It is important to highlight that, given that the constructs are different, all correlations were low or moderate, which stresses the differences between constructs despite their being related. Furthermore, data shows that attitudes towards HIV are negatively related to HIV stigma and fear of disease but are positively correlated with the knowledge of HIV and other STIs. This could indicate the importance of creating sexual health promotion programs in Colombia that are oriented towards increasing knowledge. However, these strategies cannot promote fear and reinforce the stigma around STIs. Sexual health promotion programs should focus on encouraging open and honest conversations around sexual health with a sex-positive approach to relationships.

As for the limitations of the present study, we found that even though a proportion of 20 to 1 between the number of variables and the number of participants was maintained, and therefore the design met requirements for confirmatory factor analysis [43], the sample was not representative of populations from every region and culture in the country. Therefore, future studies should sample across different parts of Colombia with diverse sociocultural profiles with the aim of reflecting the reality throughout the country in the least biased way possible. Additionally, caution interpreting scale results is recommended, especially results from Factors 3 and 4, since the reliability coefficient is not very high for confirmatory purposes [44].

## 5. Conclusions

This study presents an HIV-AS version that has been culturally adapted to Colombian populations. The scale is composed of 11 items (see Appendix A), it was validated using a thorough assessment process, and it presents satisfactory psychometric properties. This version of the test will be of great use in evaluating sexual risk factors and in assessing the effectiveness of sexual health promotion interventions, especially when focusing on adolescents. The present study also sought to contribute to the study of attitudes associated with sexual risk behaviors in Colombia.

## Figures and Tables

**Figure 1 ijerph-17-04686-f001:**
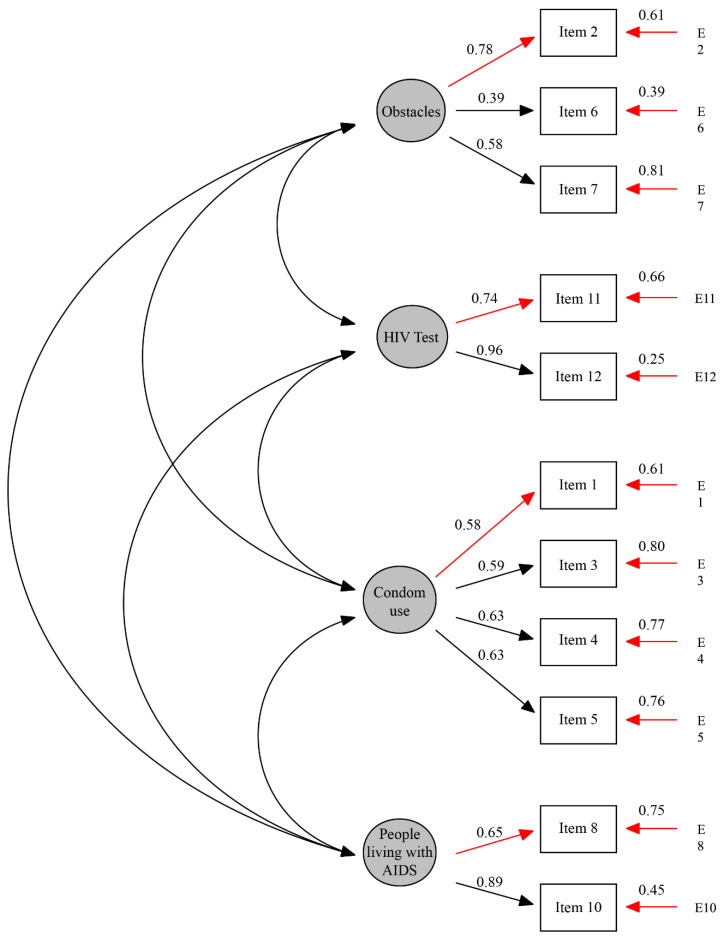
Flow chart of the model. Standardized weight.

**Table 1 ijerph-17-04686-t001:** Sociodemographic characteristics of the sample.

Sociodemographic Information	Categories	Men	Women
		^1^ M (SD) or *n* (%)	^1^ M (SD) or *n* (%)
Age		16.09 (1.43)	15.88 (1.3)
Socioeconomic level			
	Low	45 (11.4%)	73 (16%)
	Medium-Low	93 (23.5%)	105 (23%)
	Medium	228 (57.6%)	255 (55.8%)
	Medium-high	24 (6.1%)	20 (4.4%)
	High	5 (1.3%)	2 (0.4%)
Sexual orientation			
	Asexual	10 (2.6%)	6 (1.3%)
	Exclusively heterosexual	354 (90.5%)	392 (85.6%)
	2	9 (2.3%)	31 (6.8%)
	3	2 (0.5%)	11 (2.4%)
	4	8 (2.0%)	10 (2.2%)
	5	1 (0.3%)	1 (0.2%)
	6	1 (0.3%)	2 (0.4%)
	Exclusively homosexual	6 (1.5%)	5 (1.1%)
Currently has a couple			
	Yes	136(34.6%)	161 (35.5%)
	No	257 (65.4%)	292 (64.5%)
Attends religious services			
	Never	67 (16.9%)	53 (11.6%)
	Once a year	81 (20.5%)	88 (19.2%)
	Once a month	95 (24.0%)	100 (21.8%)
	At least once every three weeks	15 (3.8%)	30 (6.6%)
	At least once every two weeks	28 (7.1%)	42 (9.2%)
	At least once per week	94 (23.7%)	130 (28.4%)
	Every day	16 (4.0%)	15 (3.3%)
Sexual intercourse with penetration			
	Yes	185 (47.1%)	142 (31.1%)
	No	208 (52.9%)	314 (68.9%)

^1^ Variables are shown with absolute values and percentages.

**Table 2 ijerph-17-04686-t002:** Assessment by experts of items included in the scale of attitudes toward HIV/AIDS among adolescents.

Item	Properties	EXP. 1	EXP.2	EXP.3	EXP.4	EXP.5	M	Aiken’s V	% of Agreement	95%	
										LL	UL
	R	4	4	4	4	4	4	1		0.79	1
	P								100 %		
Item 1	C	4	4	4	4	4	4	1		0.79	1
	I	4	4	4	4	4	4	1		0.79	1
	CL	4	4	4	4	3	3.8	0.93		0.70	0.98
	R	3	4	4	3	4	3.6	0.86		0.62	0.96
	P								100 %		
Item 2	C	3	4	4	3	3	3.4	0.8		0.54	0.92
	I	4	4	4	3	4	3.8	0.93		0.70	0.98
	CL	2	4	4	3	3	3.2	0.73		0.48	0.89
	R	4	4	4	3	3	3.6	0.86		0.62	0.96
	P								100 %		
Item 3	C	4	4	4	4	3	3.8	0.93		0.70	0.98
	I	4	4	4	4	3	3.8	0.93		0.70	0.98
	CL	4	4	4	4	4	4	1		0.79	1
	R	4	4	4	3	3	3.6	0.86		0.62	0.96
	P								100 %		
Item 4	C	4	4	4	4	4	4	1		0.79	1
	I	4	4	4	4	4	4	1		0.79	1
	CL	4	4	4	4	4	4	1		0.79	1
	R	4	4	4	3	3	3.6	0.86		0.62	0.96
	P								100 %		
Item 5	C	4	4	3	4	2	3.4	0.8		0.54	0.92
	I	4	4	4	4	3	3.8	0.93		0.70	0.98
	CL	4	4	3	4	2	3.4	0.8		0.54	0.92
	R	4	4	4	3	3	3.6	0.86		0.62	0.96
	P								60%		
Item 6	C	4	4	4	4	3	3.8	0.93		0.70	0.98
	I	4	4	4	4	3	3.8	0.93		0.70	0.98
	CL	3	4	4	4	3	3.6	0.86		0.62	0.96
	R	4	4	4	4	4	4	1		0.79	1
	P								80%		
Item 7	C	4	4	4	4	4	4	1		0.79	1
	I	4	4	4	4	4	4	1		0.79	1
	CL	4	4	4	3	4	3.8	0.93		0.70	0.98
	R	4	4	4	4	4	4	1		0.79	1
	P								100 %		
Item 8	C	4	4	4	4	4	4	1		0.79	1
	I	4	4	4	4	4	4	1		0.79	1
	CL	4	4	4	4	3	3.8	0.93		0.70	0.98
	R	4	4	4	4	3	3.8	0.93		0.70	0.98
	P								100 %		
Item 9	C	4	4	4	4	2	3.6	0.86		0.62	0.96
	I	4	4	4	4	4	4	1		0.79	1
	CL	4	4	4	3	4	3.8	0.93		0.70	0.98
	R	4	4	4	4	4	4	1		0.79	1
	P								100 %		
Item 10	C	4	4	4	4	4	4	1		0.79	1
	I	4	4	4	4	4	4	1		0.79	1
	CL	4	4	4	4	3	3.8	0.93		0.70	0.98
	R	4	4	4	4	4	4	1		0.79	1
	P								100 %		
Item 11	C	4	4	4	4	4	4	1		0.79	1
	I	4	4	4	4	4	4	1		0.79	1
	CL	4	4	4	4	4	4	1		0.79	1
	R	4	4	4	4	4	4	1		0.79	1
	P								100 %		
Item 12	C	4	4	4	4	4	4	1		0.79	1
	I	4	4	4	4	4	4	1		0.79	1
	CL	4	4	4	4	4	4	1		0.79	1

Note: R, representativeness; C, comprehension; P, pertinence; I, interpretation; CL, clarity; EXP, expert; M, mean; LL, lower limit; UL, upper limit.

**Table 3 ijerph-17-04686-t003:** Component rotation matrix, communalities (h ^2^).

Item	Obstacles	HIV Test	Use of Condoms	People	h2
Item 1			0.35		0.45
Item 2	0.70				0.54
Item 3			0.50		0.34
Item 4			0.82		0.72
Item 5			0.57		0.52
Item 6	0.67				0.61
Item 7	0.77				0.61
Item 8				0.47	0.24
Item 10				0.99	1
Item 11		0.99			1
Item 12		0.59			0.45
Variance	0.36	0.13	0.11	0.09	

Factor loads lower than 0.30 were eliminated; item 9 was eliminated.

**Table 4 ijerph-17-04686-t004:** Psychometric properties of items included in the HIV-AS subscales.

Subscale	Item	M	SD	*r* _it_ ^c^	α-i	α	M (SD) Total
Obstacles	Item 2	3.01	0.90	0.53	0.60	0.73	
	Item 6	3.47	0.72	0.51	0.67		9.41 (2.64)
	Item 7	2.93	0.94	0.54	0.53		
HIV Test	Item 11	3.48	0.74	0.54	*-*	0.68	6.94 (1.26)
	Item 12	3.46	0.69	0.54	*-*		
Condom use	Item 1	3.62	0.63	0.38	0.44	0.57	
	Item 3	3.46	0.66	0.44	0.46		13.54 (2.05)
	Item 4	3.33	0.78	0.49	0.28		
	Item 5	3.14	0.78	0.49	0.44		
People Living with AIDS	Item 8	2.99	0.90	0.44	-	0.62	6.09 (3.23)
	Item 10	3.32	0.78	0.44	-		

Note. M, mean; SD, standard deviation; *r*_it_
^c^, corrected item-total correlation; α-i, ordinal alpha if item is removed; α:, ordinal alpha; item 9 was eliminated.

**Table 5 ijerph-17-04686-t005:** Matrix of correlations among factors in the HIV-AS test and other similar measures.

Subscales	MAPS−Fear of Infection	MAPS−Fear of Other People	SHAI−Negative Consequences	SHAI−Fear of the Disease	KSI−Overall Know	KSI−Other STIs	KSI−HIV Transmission	KSI−Condom	KSI−HIV Prevention
Obstacles	−0.05	−0.15 **	−0.08 *	−0.05	0.11 **	0.02	0.11 **	0.07 *	0.04
HIV Test	−0.06	−0.13 *	−0.01	0.05	0.05	0.02	0.13 **	0.04	0.07 *
Condom Use	−0.09 **	−0.12 **	−0.07 *	0.02	0.11**	0.12 **	0.17 **	0.10 **	0.20 **
People HIV	−0.12 **	−0.35 **	−0.09 **	−0.00	0.29 **	0.11 **	0.12 **	0.07 *	0.15 **

MAPS, Multicomponent AIDS Phobia Scale; SHAI, Short Health Anxiety Inventory; KSI, HIV and Other Sexually Transmitted Infections Knowledge Scale. Correlations among MAPS and SHAI subscales and knowledge about HIV and other STIs scale. (KSI); ** = p < 0.05; ** = p < 0.01.*

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
