# Peer review of "Psychometric Properties of the Colombian Version of the HIV Attitudes Scale for Adolescents"

_ijerph, 2020, doi:10.3390/ijerph17134686_

Round 1
Reviewer 1 Report
Important to validate an instrument that can effectively assess attitudes among the young population. Their attitudes influence what they do and thus the trajectory of HIV infection or their embracing of preventions.
Line 50. Define TRA
Author Response
Point 1: Line 50. Define TR
Response: We appreciate your feedback. We were using the wrong abbreviation. The right acronym is 'TBR' that stands for 'Theory of Planned Behaviour' (line 46).
Reviewer 2 Report
I suggest to propose a more explicit discussion about the interesting obtained results from tests to the real condition in Colombia: are the results meaningful? are they representative of the real situation?
Of course this depends on the kind of young people interviewed.
Author Response
Point 1: Suggest to propose a more explicit discussion about the interesting obtained results from tests to the real condition in Colombia: are the results meaningful? are they representative of the real situation?
Response: We appreciate your input. We have made sure to include these recommendations in the discussion section at 295-391 and 302-309 lines.
Reviewer 3 Report
Line 30. UNAIDS-spell it out
Line 38. Columbia had-rates-among the general population
Line 42. population. These figures a pressing need.
Line 50. TRAS spell out.
Line 97. change couple to partner
Line 131. Change title to degree
Line 137. Change other to another
Line 139. Change representativeness to representation.
Clarify MAPS, SHAI, ECI
Author Response
General response: We find your appreciation opportune. We have included it.
Point 1: Line 30. UNAIDS-spell it out:
Response: Done
Point 2: Line 38. Colombia had-rates-among the general population.
Response: Done
Point 3: Line 42. population. These figures a pressing need.
Response: Done
Point 4: Line 50. TRAS spell out.
Response: Done
Point 5: Line 97. change couple to partner
Response: Done
Point 6: Line 131. Change title to degree.
Response: Done
Point 7: Line 137. Change other to another
Response: Done
Point 8: Line 139. Change representativeness to representation.
Response: We prefer the word “representativeness” since is used by Vallejo-Medina et al., (2017), in the Guidelines for Adapting Questionnaires, to define: Ownership of the item to the sexual assertiveness construct.
Point 9: Clarify MAPS, SHAI, ECI:
Response: Done